# Isolation of Human Osteoblast Cells Capable for Mineralization and Synthetizing Bone-Related Proteins In Vitro from Adult Bone

**DOI:** 10.3390/cells11213356

**Published:** 2022-10-24

**Authors:** Daria Kostina, Arseniy Lobov, Polina Klausen, Vitaly Karelkin, Rashid Tikhilov, Svetlana Bozhkova, Andrey Sereda, Nadezhda Ryumina, Natella Enukashvily, Anna Malashicheva

**Affiliations:** 1Institute of Cytology, Russian Academy of Science, Tikhoretskiy Prospect 4, 194064 St. Petersburg, Russia; 2Vreden National Medical Research Center of Traumatology and Orthopedics, National Medical Research Center, Street Academician Baykova, 8, 195427 St. Petersburg, Russia; 3Stem Cell Bank Pokrovsky, Bolshoi Prospect V.O., 85, 199106 St. Petersburg, Russia

**Keywords:** osteoblasts, osteogenic differentiation, primary cell culture

## Abstract

The culture of osteoblasts (OB) of human origin is a useful experimental model in studying bone biology, osteogenic differentiation, functions of bone proteins, oncological processes in bone tissue, testing drugs against bone desires, and many other fields. The purpose of the present study is to share a workflow that has established the conditions to efficiently isolate and grow OB cells obtained from surgically removed bones from human donors. The protocol described here also shows how to determine cell phenotype. Here we provide characteristics of cells isolated by this protocol that might help researchers to decide if such OB are suitable for the purposes of their study. Osteoblasts isolated from collagenase-treated explants of adult bones are able to proliferate and keep their phenotype in culture. OB cells have high synthetic properties. They express osteomarkers, such as RUNX2, osteocalcin, BMP2, and osteopontin both in control conditions and in an osteogenic medium that could be estimated by qPCR and immunocytochemical staining and by Western blotting. Induction of osteogenic differentiation does not dramatically influence the synthetic properties of OB cells, while the cells gain the ability to extracellular mineralization only in an osteogenic medium.

## 1. Introduction

Osteoblasts (OB) are active bone-forming cells of mesenchyme origin, which play an important role during skeletal development and remodeling [1]. Human OB cell culture is the most relevant cell model for testing bone prostheses materials and drugs, for example, against osteoporosis. OB cells have proliferative, synthetic, endocrine, and mineralizing abilities, that can vary according to the stage of differentiation [2]. The functional characteristics of these differently staged cells forming the OB population of adult bone are at present not well understood [3]. Functional properties of isolated OB may also vary and be dependent on methods of cell isolation and cell culturing [4]. Here we present a method of OB cell isolation and the functional characteristics of cell cultures obtained by this method.

Human cells of the osteoblastic lineage can be derived from bone marrow [5], explant culture of adult bone and by osteogenic differentiation from iPSCs (induced pluripotent cells) [6]. Osteoblasts-like cell lines are also widely spread as a research model [7]. Not compromising the importance of other cell models, we suppose that explant culture is the most relevant model of adult bone. The main limitation of all cell lines derived from osteosarcomas is that some characteristics of these cells are related to their oncogenic potential rather than to osteoblastic properties. Bone-marrow-derived cells contain not only osteoblasts precursors, but could differentiate into other cell types. To avoid contamination by precursors of non-osteogenic lineage, magnet sorting with specific antibodies, such as antibodies to ALP (alkaline phosphatase) could be used. Nevertheless, direct comparisons of cells isolated from the collagenase digests of bone were clearly different from analogous cells obtained from bone marrow. Cells of bone marrow origin expressed markedly low mRNA levels of pro-osteogenic genes, although in both cases only ALP+/CD45/34/31−cells were analyzed [8].

To isolate OB cells from an adult bone two approaches could be used: isolation from untreated bone explants and osteoblast explant cultures from the collagenase-treated bone. After the treatment with collagenase, all remnants of bone marrow are released from bone pieces and could be washed away with PBS, while if untreated bones were used, cell cultures contained both bone and bone marrow–derived cells [9]. That is why we believe that treatment with collagenase is a necessary step.

Another practical question is the choice of basal cultural medium and supplements. In different studies, αMEM (Minimum Essential Medium-α) [8,9] or DMEM (Dulbecco’s Modified Eagle Medium) supplied with ascorbic acid were used. L-ascorbic acid is an important supplement both at the stage of OB isolation and culture as it is essential for normal synthesis and secretion of collagen by osteoblast cells. Moreover, the omission of the ascorbate in primary culture cannot be compensated by an addition of ascorbate to secondary culture, even for extended periods [10].

## 2. Materials and Methods

### 2.1. Patients

Samples of the femur were harvested during surgery at Vreden National Medical Research Center of Traumatology and Orthopedics. Patients were men (71.5%) and women (28.5%) from 32 to 74 years old with a median age of 52.5 years old. Patients with osteoporosis were excluded.

### 2.2. Protocol of Human OB Cells Isolation from Femur

Osteoblast-like cells were isolated from epiphysis fragments of femur bone spongy tissue. The above protocol outlines the steps necessary for the isolation and establishment of viable cell lines from human bone tissue.

#### 2.2.1. Reagents and Instruments Preparation

(All additional information about the reagents, including catalog numbers, is available in Appendix A.)

Autoclave the following items: tissue forceps, steel nippers, scalpel blade, and 0.5 L of PBS (potassium buffer solution). Add to PBS solution (Biolot, Russia) of penicillin/streptomycin (Gibco BRL, Invitrogen, Grand Island, NY, US) in the concentration recommended by the producer.Prepare collagenase type II (Worthington Biochemical Corporation, USA): solve collagenase in the volume of DMEM (Gibco BRL, Invitrogen, Grand Island, NY, US) enough to cover the piece of bone (6 mL should be enough for 1 piece of bone at about 3 × 3 cm^2^), and the final concentration of collagenase II should be 2 mg/mL. Mix well and sterilize the solution by passing through a 0.45 μm filter, then add penicillin/streptomycin in the concentration recommended by the producer. Prepare the solution just before use. Prewarm the solution to 37 °C before use.Prepare collagenase type IV the same way as collagenase type II. Final concentration of collagenase IV (Worthington Biochemical Corporation, USA) should be 2 mg/mL. Prewarm the solution to 37 °C before use.Prepare 50 mM (1000x) Ascorbic Acid (Sigma Aldrich, St. Louis, MO, USA) solution: solve Ascorbic Acid in water, filter through a 0.45 μm filter and store at −20 in aliquots.OB growth medium: DMEM base medium supplemented with 4.5 g/L glucose 2 mM L-glutamine, sodium pyruvate and penicillin/streptomycin (100 mg/L) (Gibco BRL, Invitrogen, Grand Island, NY, US), 15% FBS (fetal bovine serum, Hy Clone, Cytiva, USA), Ascorbic Acid (0.05 mM). Warm to 37 °C just before use.

#### 2.2.2. Osteoblasts Cells (OB) Isolation

A schematic representation outlining the necessary steps for the isolation, maintenance, characterization, and cryopreservation of the OB cells from the femur bone is provided in Figure 1.

All manipulations with tissue should be performed under a sterile cell culture hood at room temperature.

Place a piece of bone in a new 50 mL falcon and wash with PBS supplied with an antibiotic.Place the piece of bone in a Petri dish and add PBS with an antibiotic. Remove extraneous soft connective tissue from the outer surfaces of the bone by scraping it with a sterile scalpel blade. Change the PBS solution to avoid contamination with blood cells.Cut the fragment of bone into small (not bigger than 2 mm^2^) pieces with steel nippers. Change the PBS solution as many times as it is needed to wash the small bone pieces from blood and fat.Using the tissue forceps, place the small bone pieces into a new 50 mL falcon and add 6 mL of collagenase type II solution. Incubate for 30 min at 37 °C.Remove collagenase type II solution by serological pipette and add 6 mL of collagenase type IV solution. Incubate overnight at 37 °C. After treatment with collagenase II only, it is almost impossible to remove the remnants of blood cells from the bone tissue, while step-by-step treatment with 2 different collagenases makes it feasible.Remove collagenase type IV solution by serological pipette and wash the resulting small bone pieces with PBS solution. After collagenase treatment, the pieces will become macroscopically white as all the remnants of bone marrow will be completely removed. Then inactivate the collagenase by washing with 5–6 mL of growth culture medium, supplied with 2–10% of FBS.Scrape the cultural flask using tissue forceps. Make 4–6 scrapes. Scrapes are needed for anchoring the pieces of bone on the flask surface. It is sufficient for cell attachment and migration from explants. At the same time, it is necessary to leave enough space for intact plastic for cell growth.Add 12 mL of prewarmed cultural medium (37 °C) to the T-75 flask, place the bone pieces inside and distribute evenly on the surface of the flask.Place to CO_2_ incubator and do not move the flask for a week to avoid shift of bone explants.At day 7 check if cells are visible and change the growth medium. Then change the medium twice a week until the cells become confluent. A representative picture of OB cell isolation is represented in Figure 2.

#### 2.2.3. Osteoblasts Cells (OB) Expansion and Storage

When OB cells reach confluency of more than 90%, wash them twice with PBS to remove the excess media and detach the cells by adding the appropriate volume of pre-warmed 5x trypsin reagent (prepare from 10x Gibco Trypsin by solving with PBS) to cover the flask surface. Incubate at 37 °C for 3–7 min. After detaching the cells, inactivate trypsin with a culture medium. Centrifuge cells suspension at 180× *g* for 5 min to remove trypsin.Seed the cells on a gelatin-coated flask (0.02% gelatin solved in PBS) or Petri dish at a density of 1200–1500 cells per mm^2^. Avoid rare density as OB stop proliferating and change their morphology without cell-to-cell contacts. It is recommended to seed OB cells 1:1 in passage 1 and 1:2 or 1:3 in the next passages. OB cells grow rather slowly and rich confluence in a week. Use the cells between passages 2–5.Cryoconservation: Remove cells from the Petri dish as described in step 1. After trypsinization and centrifugation resuspend the cell pellet in the previously prepared mix: 90% of FBS plus 10% of DMSO (Dimethyl sulfoxide Sigma Aldrich, St. Louis, MO, USA) to a cell density of ~1 million cells per 1 mL. Add each ml of the cell suspension to a cryopreservation vial. Place vials in a cell-freezing container. Place the container at −80 °C for 6–72 h or according to the freezing container protocol. Remove vials from −80 °C and transfer to liquid nitrogen for long-term storage.For all experiments cells passages between 2–5 are used. After passage 5 the cells usually lose osteogenic potential.

### 2.3. Osteogenic Differentiation

To induce osteogenic differentiation OB cells were seeded at 80% confluency in a growth medium. G The following number of cells were used: 200 × 10^3^ cells per well for 6-well plate, 100 × 10^3^ for 12-well plate, and 60 × 10^3^ for 24-well plate. After 24 h the medium was changed to osteogenic media [11,12] containing high-glucose nutrient medium DMEM 4.5 supplemented with 10% FBS, 1% penicillin/streptomycin, 1% glutamine 10 mM β-glycerolphosphate, 200 μM L–Ascorbic acid and 100 nM dexamethasone (all Sigma Aldrich, St. Louis, MO, USA). The culture medium was changed every 3 days. Fresh factors of osteogenic differentiation were added to the cultural medium just before use. Immunocytochemical staining was performed and gene expression was analyzed 96 h after the induction of differentiation. Alizarin staining was performed 21 days after induction of differentiation.

### 2.4. Western blotting

The OB cells were seeded in gelatin-coated 6-well plates at a density of 200 × 10^3^ cells per well in standard conditions for 96 h. The cells were treated with 150 μL of RIPA Buffer (Sigma Aldrich, St. Louis, MO, USA) and a protease inhibitor cocktail. The total protein level was measured by BSA assay kit (Thermo Scientific, Waltham, MA, USA) using Varioscan (Thermo Scientific, Waltham, MA, USA). An equal amount of protein from each donor was used for the assay. Extracts were separated by 10% sodium dodecyl sulfate-polyacrylamide gel electrophoresis (SDS-PAGE) (Bio-Rad, CA, USA). Primary antibodies for RUNX2 were used for Westernblotting detection of RUNX2 protein.

### 2.5. Reverse Transcription-PCR

Total RNA was extracted from OB cell cultures using Trizol reagent (Eurogen, Russia) according to the instructions of the manufacturer. Reverse transcription was performed using corresponding kits. Real-time PCR was performed with SYBR Green detection using specific primers in technical replicates. The mRNA levels were normalized to GAPDH mRNA. Changes in target gene expression levels were calculated as fold differences using the comparative ΔΔCT method. The primer sequence is available in Appendix B (Table A2).

### 2.6. Immunocytochemistry

For immunocytochemical staining, OB were grown on cover microscopic slides covered with gelatin in wells of 48-well plates at a density of 50 × 10^3^ cells per well either in the standard culture conditions or in osteogenic medium for 96 h. Then the cells were washed with PBS and fixed for 10 min in 4% paraformaldehyde, washed with PBS three times, and permeabilized in 0.1% Triton X (Sigma Aldrich, St. Louis, MO, USA) –100/PBS for 10 min, followed by blocking in 1% BSA (Sigma Aldrich, St. Louis, MO, USA) /PBS for 30 min. Then the cells were incubated overnight at +4 °C with primary antibodies: BMP-2/4 (Santa Cruz Biotechnology, Santa Cruz, CA, USA), RUNX2 (Abcam, UK), osteocalcin (Abcam, UK), and osteopontin (Santa Cruz Biotechnology, Santa Cruz, CA, USA) (all in dilution 1:100). The slides with the cells were washed three times with PBS and incubated with secondary antibodies conjugated with Alexa488 (Invitrogen, Carlsbad, CA, USA) in dilution 1:1000 for 1 h at room temperature. DAPI was used to visualize nuclei. Microphotographs were made using a confocal microscope 400× objective.

### 2.7. Flow Cytometry

OB were seeded at 25 cm^2^ flasks (Corning) coated with 0.02% gelatin (Sigma Aldrich, St. Louis, MO, USA) at a density of 400 × 10^3^ cells per flask. On day 4 the cells were removed and immunostained with the following panels of monoclonal antibodies: CD44-FITC/CD73-PE/CD90-PC5/CD105-PC7 and CD34-FITC/CD117-PE/CD14-PC5/CD45-PC7 (Beckman Coulter, Fullerton, CA). Analysis was performed using a flow cytometer according to standard protocols recommended by the manufacturer. The autofluorescence level was evaluated using an unstained control sample. The level of non-specific binding of antibodies was determined using isotypic controls (mouse immunoglobulins conjugated to FITC, PE, PC5, PC7). Gating of fluorescence events was carried out using the viability parameter. The viability was estimated by forward and side scattering along with 7-aminoactinomycin D staining. In each sample, at least 15,000 “targeted events” (events determined as viable cells) were analyzed.

### 2.8. Alizarin Staining

For alizarin red staining cells were seeded on and covered with gelatin wells of 48 well plates at a density of 40 × 10^3^ cells per well. Calcium deposition was detected by Alizarin Red (Sigma Aldrich, St. Louis, MO, USA) to staining at day 21 of differentiation. OB cells on wells were washed with PBS twice, fixed with 70% ethanol for 1 h, then washed with distilled water twice, and incubated with Alizarin red solution for 40 min. Alizarin red solution had a dilution of 11.52 mg/mL. Then the wells were twice washed with distilled water and scanned using a scanner (Arcus Agfa 1200, Mortsel, Belgium).

### 2.9. Statistical Analysis

For flow cytometry assay values are expressed as median ± standard deviation. For reverse transcription-PCR values are expressed as median ± range. Groups were compared using the Wilcoxon paired nonparametric one-tailed *t*-test. A value of *p* ≤ 0.05 was considered significant.

## 3. Results

### 3.1. OB Cell Cultures Characteristics

We have studied characteristics of the OB cells from 5 donors at passage 2 and at passage 4 by flow cytometry (Table A1). To check the cell viability of OB we have stained OB with PI (propidium iodide). OB demonstrated high cell viability both at passage 2 (90 ± 3.4%) and passage 3–4 (91.3 ± 0.9%). We detected a subpopulation of bigger cells with high autofluorescence in all OB cell cultures (37% ± 12.2%) at passage 2, while at passage 3–4 the amount of this subpopulation decreased (13% ± 7%). All OB cultures were positive for CD44, CD73, CD90, and CD105 (95–100% positive cells) and negative for CD14, CD34, CD45, CD117, and HLA-DR that characterizes described OB as a mesenchymal cell line [13] and confirms their origin from bone tissue, but not from blood. OB cell population was rather homogenous and stable from passage to passage (Table 1).

To confirm an osteoblastic phenotype of the cells, we analyzed their ability to express osteogenic markers by Western blotting, qPCR, and immunocytochemical staining.

We have confirmed by Western blotting that OB cells isolated by described method contain marker protein of osteoblasts, RUNX2 (Figure 3).

### 3.2. Comparison of OB Cells Properties in Control Conditions and in Osteogenic Medium

According to alizarin staining, OB cells are able to differentiate and deposit calcium in an osteogenic medium (Figure 4).

According to the qPCR assay, isolated OB cells express their specific markers both in control conditions and in an osteogenic medium (Figure 5). They express *BGLAP* (bone gamma-carboxyglutamate protein), which encodes osteocalcin, the most abundant protein in bone [14], *RUNX2*, the osteoblasts master regulator [15], and *SPP1*, which encodes osteopontin (*OPN*), the protein, that is known to be expressed in bone and bone marrow by osteoblastic and osteoclastic precursor cells, multi-nucleated osteoclasts, osteocytes, and osteoblasts [16], *ENPP1*, encoding ectonucleotide pyrophosphatase/phosphodiesterase which helps to regulate bone and cartilage mineralization by producing inorganic pyrophosphate [17], *PERIOSTIN*, *BMP2*, and *COL1A*, which encodes type 1 collagen. OB cells also express *ACTA2*, a marker gene for mesenchymal cells. *ENPP1*, *BGLAP*, and its regulator *RUNX2* were upregulated in OM, while *PERIOSTIN*, *BMP2*, and *COL1A1* were downregulated in OM. We have not found significant changes in *SPP1* and *ACTA2* levels in response to osteogenic differentiation, although there was a tendency toward an upregulation of *SPP1* in OM.

To verify if OB cells really have an osteogenic phenotype, we performed immunocytochemical staining to OPN (osteopontin), RUNX2, BMP2/4, and osteocalcin. OB cells do express RUNX2, osteopontin, BMP2/4, and osteocalcin proteins in control conditions. After cultivation in an osteogenic medium for 96 h, we observed an increase in the level of osteocalcin, a slight increase in the level of osteopontin, and a decrease in the level of BMP2/4 by immunocytochemical staining (Figure 6).

## 4. Discussion

According to flow cytometry characteristics, OB cell cultures are stable at passages 2–4 and their cell viability is high. These cells have a proliferative activity, and could be frozen and cryopreserved, and thawed without visible loss of their properties. This characterizes them as a suitable research model. The advantage of the cells, isolated by the described method compared to bone marrow-derived osteoprogenitors, is that the resulting cell population is more homogenous. According to flow cytometry characteristics, there is no contamination of blood-derived cells in OB cultures and one does not need to use sorting after cell isolation, while for bone marrow-derived osteoblasts it is a necessary step [18]. This makes the described method cheaper and faster. Besides, in bone marrow-derived osteoblasts, cells of blood lineage and adipocyte precursors are present [5]. We have not observed any signs of the presence of adipocytes or adipogenic differentiation in our OB cultures. Their differentiation abilities and basal level of osteogenic markers production also varied from donor to donor, while the direction of changes was similar in all cultures in our experiments. According to alizarin staining, OB cells have high ability to mineral deposition in osteogenic medium and even express osteogenic markers in standard conditions as shown by PCR, WB, and immunocytochemical staining. We also observed by light microscopy that OB cells synthesized extracellular matrix granules in culture in standard conditions (Figure 2). Using the described method of cell isolation, OB cells with predicted characteristics and high osteogenic differentiative abilities could be obtained. In a recent study, a method of osteoclast isolation from the same source as ours (human femur) has been described [19]. This highlights that different protocols could lead to receiving cells with different phenotypes. The main limitations of explant cultures of OB include a long time of cell isolation as well as the slow proliferation of cell cultures. Primary OB cultures are slightly heterogenic. To obtain a more homogenous cell population antibody sorting could be used after the cell isolation. The properties of osteoblasts isolated from different parts of the skeleton are not similar [20]. Here we studied OB cells derived from femur bone and some of their properties could be unique for this type of bone.

The role of bone-related proteins in mineralization relates unclear [21]. In our cell model, isolated osteoblast cells have high synthetic properties, but they gain the ability to extracellular mineralization only in an osteogenic medium. An increase of *RUNX2*, *BGLAP*, and *ENPP1* mRNA levels was observed in response to the stimulation of osteogenic differentiation. We also observed an increase in osteocalcin and osteopontin protein content. We observed that relative mRNA levels of *COL1A1*, *BMP2*, and *POSTIN* decreased in OB cells after initial culture in an osteogenic medium. All these markers are associated with osteogenic differentiation and bone formation, but are not unique to bone tissue and take part in many other biological processes. Periostin (*PERIOSTIN*) regulates cell adhesion, and differentiation and plays a role in transducing mechanical signals both in bone and non-bone tissues, that are exposed to mechanical stress, including the periodontal ligament and heart valves [22]. Periostin also plays an important role in fibrosis and cancer [23]. Osteopontin (*SPP1*) has long been believed to function as an inhibitor of the calcification process. It is required for osteoclastic resorption of bone. Neither osteopontin, nor periostin and osteocalcin null mice are lethal or display major bone abnormalities [16]. Type 1 collagen is a key marker of an early proliferation and synthetic phase of osteogenic differentiation [18]. The requirement of collagen, osteopontin, and periostin at the stage of mineralization may be decreased.

The paper where the isolation of osteoblasts from bones had been described was published in 1977 [24]. Nevertheless, we suggest the importance and relevance of our study, as in early papers the cells were not well-characterized, protocols did not contain details, and were not reproducible. It is important to update research protocols and provide cell characterization using contemporary methods for primary cell cultures. We speculate that the bone origin of the cells and positive alizarin staining together confirm that OB isolated by the described method is a relevant model of normal osteogenic differentiation. It is known that differentiation processes are dependent on cell and tissue specificity. The difference in osteogenic differentiation between smooth muscle cells and osteoblasts has been shown in murine cells [25]. Osteogenic differentiation of MSC of non-bone origin is actually rather related to pathological than to physiological processes which take place during bone formation. In our previous works, we have shown that the osteogenic differentiation of different types of mesenchymal cells demonstrates some variability in response to osteogenic stimuli [26]. In vascular and valvular calcification, a response visible by expression of osteogenic markers to osteogenic stimuli may vary not only in different cell types, but also might be dependent on the type of pathological process [27,28]. Moreover, it is known that the mechanisms of osteogenic differentiation vary between species and a response of osteoblasts of non-human origin to pharmacological agents may differ from that of human osteoblasts. In our recent review, we discussed that many alterations are described for the expression of transcription factor RUNX2 in response to osteogenic stimuli and thus the expression of RUNX2 might be cell type/model-dependent [18]. For example, dexamethasone stimulated RUNX2 production in human osteoblast cell lines, but inhibited RUNX2 in rodent osteoblasts [18,29]. That is the reason to test antiosteoporosis drugs and materials for bone prosthesis using human osteoblasts of bone origin in preclinical studies in addition to animal models. The novelty of this research is that we suggest a panel of reliable osteogenic markers that match normal osteogenic changes and could be used as a reference for studying pathological calcification, bone formation, and osteoporosis. We have shown that donor to donor effect is quite prominent and the level of production of osteomarkers varies both in normal culture conditions and after osteogenic stimulation. We suggest paying attention to this fact while planning experiments with osteoblasts. We have shown that 96 h is enough for detecting osteogenic changes at the mRNA level and consider this time point as a reliable estimation of osteogenic changes.

Together, our findings characterize OB as cells at late differentiation stage with high synthetic abilities, ability to proliferation, and extracellular mineralization. Here we have shown how the expression of some proteins depends on osteogenic differentiation in mature osteoblasts. It is important for understanding the nature of the mineralization process and for the estimation of the role of these proteins in bone metabolism.

## Figures and Tables

**Figure 1 cells-11-03356-f001:**
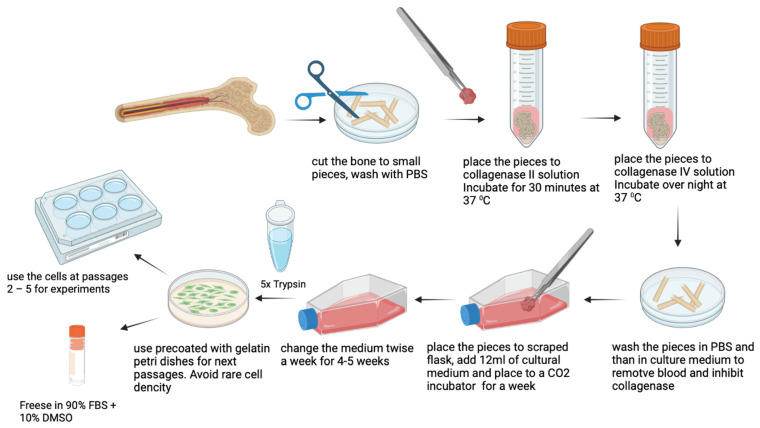
**Schematic representation outlining the necessary steps for the isolation,** **maintenance, characterization, and cryopreservation of the OB cells.**

**Figure 2 cells-11-03356-f002:**
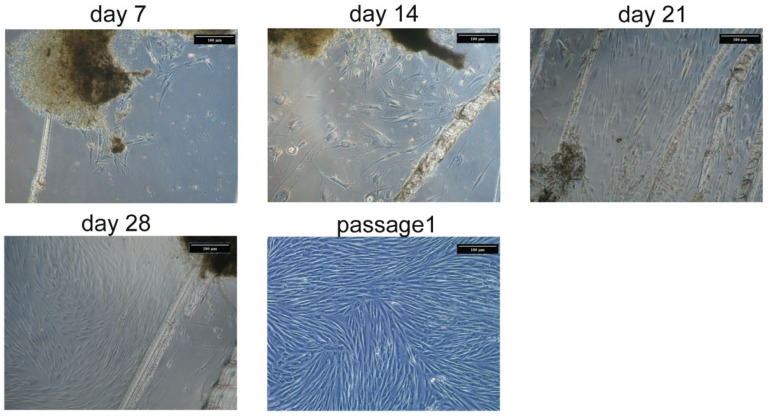
**Representative photos of cell isolation from bone explants.** Scrapes on the culture flask surface help to fix bone pieces. First cells were observed at 3–7 days of isolation. It takes 4–5 weeks before cells reach confluence and could be passaged. After passaging cell culture becomes more homogenous.

**Figure 3 cells-11-03356-f003:**
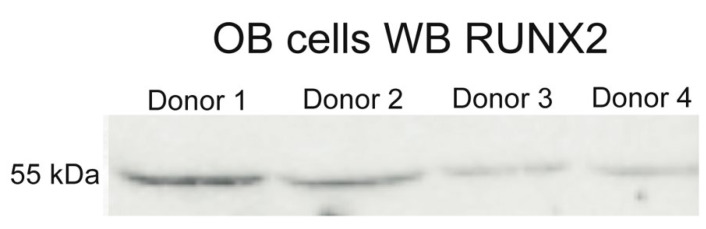
**OB cells express RUNX2,** **marker protein of osteoblasts.**

**Figure 4 cells-11-03356-f004:**
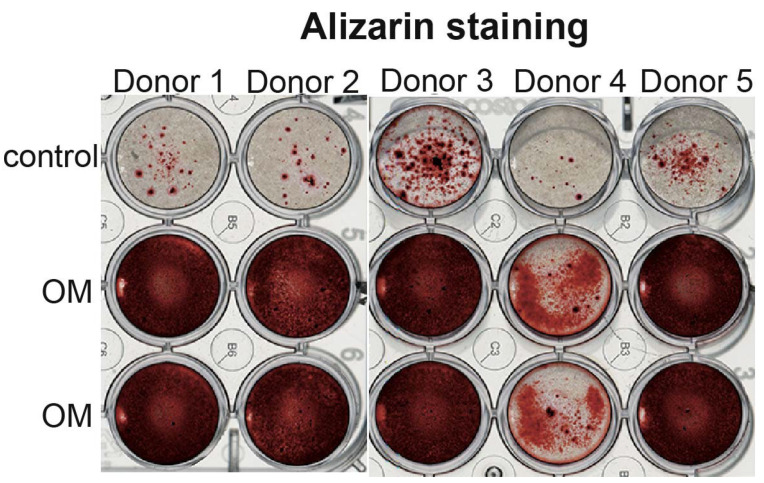
**OB cells are able to osteo differentiation.** Alizarin staining of OB cells in the control condition and on the 21st day of differentiation in osteogenic medium (OM).

**Figure 5 cells-11-03356-f005:**
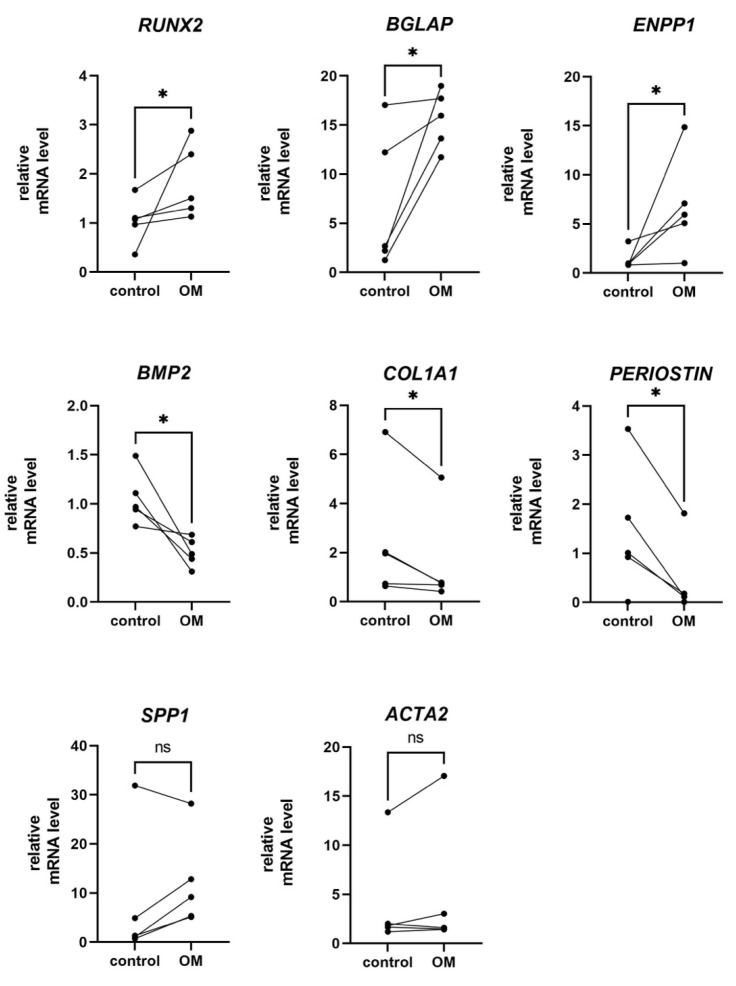
**OB cells phenotype characterization by qPCR in control conditions and 96 h of differentiation in osteogenic medium (OM).** OB cells from all tested donors showed a similar response to osteogenic medium, while relative mRNA levels both in control and experimental conditions vary from donor to donor. * *p* ≤ 0.05, ns-non significant.

**Figure 6 cells-11-03356-f006:**
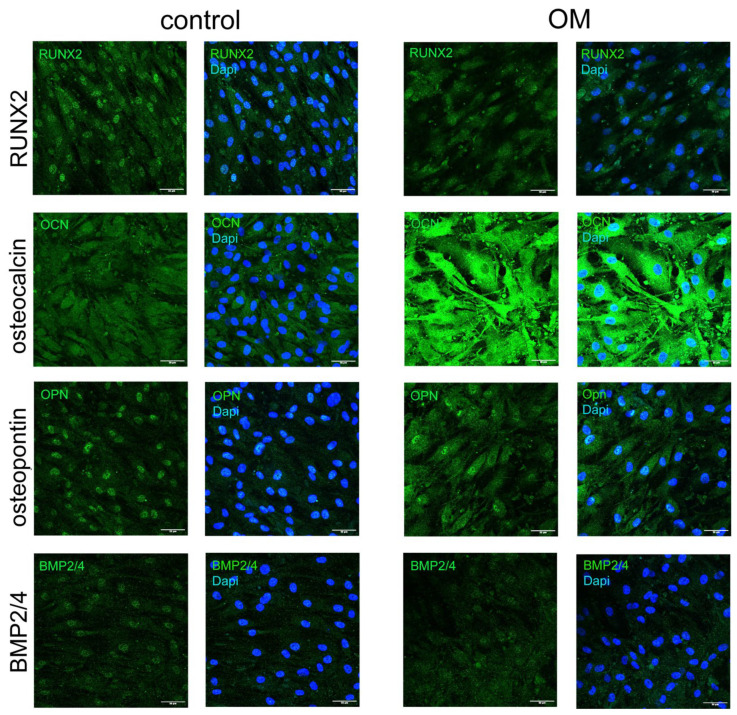
**Immunocytochemical staining of OB cells shows that these cells express osteomarkers both in control and differentiative conditions.** Osteo medium (OM) was added to 96 h. Osteocalcin was upregulated in differentiative conditions.

**Table 1 cells-11-03356-t001:** **Characterization of OB cell size range, cell viability, and surface marker expression by flow cytometry.** OB cells are rather heterogenic, have high cell viability, and are positive to mesenchymal cell markers and negative to markers of blood lineage. We have not detected significant changes in the characteristics of OB cells between passages 2–4. P-passage.

Donor	Bigger Cell Size,%	Cell Viability,%	CD 90,%	CD 105,%	CD 73,%	CD 44,%	CD14, CD45, CD34, CD117, HLA-DR
Donor 1	P2-12	P2-95.2	P2-99	P2-100	P2-99.8	P2-99.2	All negative
	P4–13	P4-91.3	P4-99.5	P4-100	P4-99.5	P4-98.6	
Donor 2	P2-38	P2-86.8	P2-98.5	P2-99.7	P2-98.1	P2-98.7	All negative
	P4-22	P4-89.9	P4-99	P4-99.7	P4-99.2	P4-99.1	
Donor 3	P2-18	P2-89.8	P2-97.4	P2-99.9	P2-99.2	P2-99.6	All negative
	P4-5	P4-91.3	P4-98.8	P4-99.9	P4-99.2	P4-98.6	
Donor 4	P2-41	P2-89.4	P2-98.2	P2-99.6	P2-99.3	P2-99.6	All negative
Donor 5	P2-37	P2-95.3	P2-98.4	P2-99.9	P2-99.5	P2-99.7	All negative
Median ± st deviation	P2-37 ± 12	P2-90 ± 3.4	P2-98.5 ± 0.5	P2-99.9 ± 0.1	P2-99.3 ± 0.6	P2-99.6 ± 0.4	
	P4-13 ± 7	P4-91 ± 0.95	P4-99 ± 0.3	P4-99.9 ± 0.1	P4-99.2 ± 0.2	P4-98.6 ± 0.4	

## Data Availability

Not applicable.

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
