# Peer review of "Isolation of Human Osteoblast Cells Capable for Mineralization and Synthetizing Bone-Related Proteins In Vitro from Adult Bone"

_cells, 2022, doi:10.3390/cells11213356_

Round 1

Reviewer 1 Report

In this manuscript, the authors present a method to isolate and grow the human osteoblast cells obtained from surgically removed bones from donors. In the introducing section they describe in short different methods of osteoblasts isolation. Next, the authors report a detailed protocol and show the characteristics of osteoblast cell cultures obtained by this way. This manuscript is of great interest to specialists, but to a lesser extent to a wide range of readers.

The discussion should be expanded by comparing the known procedures for the isolation and cultivation of human osteoblasts and the expression of the advantages and possible disadvantages of the method developed by the authors.

Minor Comments:

Please, specify the default temperature conditions or indicate them for each procedure step.

Author Response

Reviewer 1

1.The discussion should be expanded by comparing the known procedures for the isolation and cultivation of human osteoblasts and the expression of the advantages and possible disadvantages of the method developed by the authors.

Answer: Thank you for your useful recommendation to pay attention to this important point. Discussing previously known method of osteoblasts isolation in comparison with the provided method, we would like to notice that the paper where isolation of osteoblasts from bones were performed at the first time were published in 1977 (Yagiela and Woodbury 1977). Nevertheless, we suggest an importance and relevance of our study, as in early papers cells were not well characterized, protocols did not contain details and were not reproduceable. It is important to update research protocols and provide cell characterization using contemporary methods for primary cell cultures.

An advantage of the cells, isolated by described method comparing to bone marrow derived osteoprogenitors, is that the resulting cell population is more homogenic. According to flow cytometry characteristics, there are no contamination of blood derived cells in OB cultures and it is not needed to use sorting after cell isolation, while for bone marrow derived osteoblasts it is necessary step (Fujita et al. 2014). This makes the described method cheaper and faster. Besides cells of blood lineage, in bone marrow osteoblasts and adipocytes precursors are present (Rickard et al. 2009). We haven’t observed any signs of presence of adipocytes or adipodifferentiation in described OB cultures. In a recent study a method of osteoclasts isolation from the same source as ours (human femur) has been described (Bernhardt et al. 2020). This highlights that different protocols could lead to receiving cells with different phenotypes.

  1. Please, specify the default temperature conditions or indicate them for each procedure step.

 Answer: thank you for your suggestion, we agree that this information would be useful. According to your recommendations, we have added this information to the protocol.

Reviewer 2 Report

The purpose of the manuscript by Kostina and colleagues is to provide a protocol for isolating and culturing primary human osteoblasts from collagenase-treated explants of adult bones, as well as characterize and monitor their functions. The ability to culture human bone cells is an essential step to study and fully understand the pathological mechanisms that underline bone diseases.

 Overall, the protocol seems well performed and several techniques to verify the osteoblastic phenotype are showed, although not fully properly explained and discussed. Major editing for proper use of the English language is indispensable. Several deficiencies in the presentation of the methods and writing of the protocols make it difficult to fully evaluate the scientific contribution of the article. 

Specific points:

1.      The protocol/article would benefit from a schematic representation outlining the necessary steps for the isolation, maintenance, characterization and cryopreservation of the cells isolated.

2.      Include a section with the list of all the materials necessary for isolation, culturing, passaging of secondary culture, phenotypic characterization and cryopreservation.

3.      In the materials list indicate the concentrations and volumes of the solutions needed, including notes and explanations why the authors used for example, 2 different collagenases for the digestion of the bone explants.

4.      The methods section should be divided in subsections in which every step is meticulously described.

a.       Section 2.2.2 which bones where used? Do bone fragments need to be kept under rotation during collagenase digestions?

b.      Section 2.2.3 how many cells are plated for expansion? How many passages are the primary cells cultured for? Which passage was used for the various assay? Why were cells seeded in gelatin coated flask?

c.       Section 2.3 How often the media was replaced? was the osteogenic differentiation medium prepared fresh every time.

d.      Section 2.4 How many cells were plated? how long the culture was kept for? how much RIPA was used? Add REF if the methodology is not explained

e.       Section 2.5 Please have a table with the sequence of the primers used for the rt-PCR

f.        Section 2.6 How many cells were plated?

g.      Section 2.8 How many cells were plated? Which culture plate was used? How the Alizarin Red solution was prepared, describe the technique, cite REF.

5.      Results section is not needed as this is not an original article and it does not show any novel finding. Every result should be included at the end of each method section to demonstrated which result should be obtained by following the technique described in the protocol.

6.      Discussion section should be expanded with recommendations and tips from the authors. Also, similarities and differences with the other protocols already present in literature should be discussed.

7.      Abbreviation needs to be spelled out at least the first time 

Author Response

Reviewer 2

  1. 1. The protocol/article would benefit from a schematic representation outlining the necessary steps for the isolation, maintenance, characterization and cryopreservation of the cells isolated. 

Answer: thank you very much for this advice. We have added the scheme outlining the necessary steps of cell isolation.

  1. Include a section with the list of all the materials necessary for isolation, culturing, passaging of secondary culture, phenotypic characterization and cryopreservation. 

Answer: thank you for your suggestion. We have added the table of materials to Appendix A to make the protocol easier to reproduce.

  1. In the materials list indicate the concentrations and volumes of the solutions needed, including notes and explanations why the authors used for example, 2 different collagenases for the digestion of the bone explants.

Answer: thank you for your interest to details. We have added the volume of solutions to the text. We used 2 different collagenases as bone tissue is rich of extracellular matrix, especially collagen 1. After treatment with only collagenase II, it was impossible to remove the rest of blood from the bone tissue, while step by step treatment with II collagenases made it possible.

4.The methods section should be divided in subsections in which every step is meticulously described.

Answer: thank you for your interest to details. We have added them to the protocols.

  1. Section 2.2.2 which bones where used? Do bone fragments need to be kept under rotation during collagenase digestions?

Answer:  Fragments of femur bones were used.

  1. Section 2.2.3 how many cells are plated for expansion? How many passages are the primary cells cultured for? Which passage was used for the various assay? Why were cells seeded in gelatin coated flask?

Answer: It is important to avoid rare cell density. We recommend cell density 1200-1500 cells per mm2. All experiments were performed between passages 2-5. After passage 6 cells are growing slowly. Gelatin coating is standard procedure for primary mesenchyme cell culturing and bones are rich in gelatin and gelatin coating favors cell attaching to the plates.

  1. Section 2.3 How often the media was replaced? was the osteogenic differentiation medium prepared fresh every time.

Answer: The culture medium was changed every 3 days. Fresh factors of osteogenic differentiation were added to cultural medium just before use.

  1. Section 2.4 How many cells were plated? how long the culture was kept for? how much RIPA was used? Add REF if the methodology is not explained 

Answer:  The OB cells were seeded at gelatin coated 6 well plate in amount of 200 x 103 cells per well in standard conditions for 96 hours. The cells were treated with 150 mkl of RIPA Buffer and protease inhibitor cocktail.

  1. Section 2.5 Please have a table with the sequence of the primers used for the rt-PCR

Answer:  We have added the table with primer sequence. The primer sequences is available in Appendix B

  1. Section 2.6 How many cells were plated?

Answer:  For immunocytochemical staining cells were grown on cover slides covered with gelatin in wells of 48 well plates in amount of 50 X 103 cells per well

  1. Section 2.8 How many cells were plated? Which culture plate was used? How the Alizarin Red solution was prepared, describe the technique, cite REF.

Answer:  For alizarin red staining cells were seeded on covered with gelatin wells of 48 well plates in amount of 40 X 103 cells per well.

  1. Results section is not needed as this is not an original article and it does not show any novel finding. Every result should be included at the end of each method section to demonstrated which result should be obtained by following the technique described in the protocol.

Answer:  In the result sections we show the characteristics of OB isolated by the described method. Replacing the figures to the materials and methods section would make it difficult to understand the logic of the article.

  1. Discussion section should be expanded with recommendations and tips from the authors. Also, similarities and differences with the other protocols already present in literature should be discussed.

 Answer:   Thank you very much for your recommendations how to improve our article. In the revised version of the paper we discuss previously known method of osteoblasts isolation in comparison to the provided method. We would like to notice that papers where isolation of osteoblasts from bones was performed at the first time were published in 1977 (Yagiela and Woodbury 1977). Nevertheless, we suggest an importance and relevance of our study, as in early papers cells were not well characterized, protocols did not contain details and were not reproduceable. It is important to update research protocols and provide cell characterization using contemporary methods for primary cell cultures.

Advantage of the cells, isolated by described method comparing to bone marrow derived osteoprogenitors, is that cell population is more homogenous. According to flow cytometry characteristics, there are no contamination of blood derived cells in OB cultures and it is not needed to use sorting after cell isolation, while for bone marrow derived osteoblasts it is necessary step (Fujita et al. 2014). This makes the described method cheaper and faster. Besides cells of blood lineage, in bone marrow osteoblasts and adipocytes precursors are present (Rickard et al. 2009). We haven’t observed any signs of presence of adipocytes or adipogenic differentiation in described OB cultures. In a recent study a method of osteoclasts isolation from the same source as ours (human femur) has been described (Bernhardt et al. 2020). This highlights that different protocols could lead to receiving cells with different phenotypes.

  1. Abbreviation needs to be spelled out at least the first time 

 Answer:   Thank you for your recommendation. We have added spelling of the abbreviations to the text.

Reviewer 3 Report

Comments

The aim of this study seems to share a workflow that has established the conditions to efficiently isolate and grow osteoblastic cells isolated from human-originated bones. The current study showed characteristic phenotypes of these cells under the conditions of control and osteogenic differentiation media through application of various experimental methods. In summary, the authors state that induction of osteogenic differentiation does not dramatically influence synthetic properties of osteoblastic cells, while they gain ability to extracellular mineralization only in osteogenic medium.

Although this study used bone samples derived from human donors and showed several characteristic properties to synthesize osteogenic proteins or mineralize in relation to the culture conditions, almost of data in the current study appeared to be preliminary results. There are not any novel or specific findings from the isolation of osteoblastic cells to the characterization of these cells according to the osteogenic differentiation. If the authors want to show a scientific new finding in their study, the investigation on whether or how the isolated and differentiated human-derived osteoblasts could be applied to a clinical or to preclinical approach/study might be shown in the current results. Without any trials to show some new findings or theories on bone or its cells, the manuscript couldn't be considerable for acceptance even if the cells are originated from human samples. In addition, the current manuscript is required for severe revision of English editing.

Author Response

Reviewer 3

1.Although this study used bone samples derived from human donors and showed several characteristic properties to synthesize osteogenic proteins or mineralize in relation to the culture conditions, almost of data in the current study appeared to be preliminary results. There are not any novel or specific findings from the isolation of osteoblastic cells to the characterization of these cells according to the osteogenic differentiation. If the authors want to show a scientific new finding in their study, the investigation on whether or how the isolated and differentiated human-derived osteoblasts could be applied to a clinical or to preclinical approach/study might be shown in the current results. Without any trials to show some new findings or theories on bone or its cells, the manuscript couldn't be considerable for acceptance even if the cells are originated from human samples. In addition, the current manuscript is required for severe revision of English editing.

Answer: Thank you very much for your recommendations how to improve our article. Indeed, the opportunity to apply the results is a principal point of relevance of the research. We speculate that bone origin of the cells and positive alizarin staining together confirms that OB isolated by described method is a relevant model of normal osteogenic differentiation. It is known that differentiation processes are cell and tissue specific. Difference in osteogenic differentiation between smooth muscle cells and osteoblasts was shown for murine cells (Patel et al. 2019). Osteogenic differentiation of MSC of non-bone origin is actually rather related to pathological than to physiological processes which take place during bone formation. In our previous wokks we have shown that osteogenic differentiation of different types of mesenchymal cells demonstrate some variability in response to osteogenic stimuli (Aleksandra Kostina et al. 2021). In vascular and valvular calcification a response visible by expression of osteomarkers to osteogenic stimuli may vary not only in different cell types, but also might be dependent  on the type of pathological process (Ignatieva et al. 2017), (A. Kostina et al. 2018). Moreover, it is known that the mechanisms of osteogenic differentiation vary between species and a response of osteoblasts of non-human origin to pharmacological agents may differ from that of human osteoblasts. In our recent review we discuss that many alterations are described for expression of transcription factor Runx2 in response to osteogenic stimuli and thus the expression of Runx2 might be cell type/model-dependent. For example, dexamethasone stimulated Runx2 production in human osteoblast cell lines, but inhibited Runx2 in rodent osteoblasts (Lobov and Malashicheva 2022). That is the reason to test antiosteoporosis drugs and materials for bone prothesis on human osteoblasts of bone origin in preclinical studies in addition to animal models. Novelty of this research is that we suggest a panel of reliable osteomarkers that match normal osteogenic changes that could be used as a reference for studying pathological calcification, bone formation and osteoporosis. We have shown that donor to donor effect is quite prominent and the level of production of osteomarkers vary both in normal conditions and after osteogenic stimulation. We suggest to pay attention to this fact while planning experiments with osteoblasts. We have shown that 96 hours is enough for detecting osteogenic changes at the mRNA level and consider this time point as a reliable for estimation of osteogenic changes.

Reviewer 4 Report

The manuscript describes the isolation and characterization of osteoblasts from discarded human surgical samples. The cells adhere to tissue culture plates, proliferate, and express markers of osteoblastic differentiation as assessed by mRNA and protein expression. Mineralization, assessed using Alizarin red staining, increases when the cells are cultured in osteogenic medium containing ascorbic acid and beta-glycerophosphate.

The methodology is very standard and similar protocols have been reported before. If there are any novel aspects to the method used, it has not been highlighted in the Discussion.

Specific comments:

1.  Although fairly well written, the text would benefit from proofreading by someone with better command of the English language. Some words are used based on phonetics and not on meaning (live instead of leave; grows instead of growth; rich instead of reach; etc.)

2.      The Discussion is very descriptive and brings no element of novelty.

3.    There is no discussion of how the current protocol differs from previous methods.

Author Response

Reviewer 4

  1. Although fairly well written, the text would benefit from proofreading by someone with better command of the English language. Some words are used based on phonetics and not on meaning (live instead of leave; grows instead of growth; rich instead of reach; etc.)

Answer: thank you very much for your comment. We have corrected the mistakes.

  1. The Discussion is very descriptive and brings no element of novelty.

Answer: Thank you for your recommendation. We have expanded the discussion and highlighted the aspects of novelty of the provided method of osteoblasts isolation. Discussing previously known method of osteoblasts isolation in comparison with the provided method, we would like to notice that the paper where isolation of osteoblasts from bones was performed for the first time were published in 1977 (Yagiela and Woodbury 1977). Nevertheless, we suggest the importance and the relevance of our study, as in early papers cells were not well characterized, protocols did not contain details and were not reproduceable. It is important to update research protocols and provide cell characterization using contemporary methods for primary cell cultures.

Advantage of the cells, isolated by described method comparing to bone marrow derived osteoprogenitors, is that the cell population is more homogenic. According to flow cytometry characteristics, there are no contamination of blood derived cells in OB cultures and it is not needed to use sorting after cell isolation, while for bone marrow derived osteoblasts it is necessary step (Fujita et al. 2014). This makes the described method cheaper and faster. Besides cells of blood lineage, in bone marrow osteoblasts and adipocytes precursors are present (Rickard et al. 2009). We haven’t observed any signs of presence of adipocytes or adipogenic differentiation in described OB cultures. In a recent study a method of osteoclasts isolation from the same source as ours (human femur) has been described (Bernhardt et al. 2020). This highlights that different protocols could lead to receiving cells with different phenotypes.

  1. There is no discussion of how the current protocol differs from previous methods.

Answer: We speculate that bone origin of the cells and positive alizarin staining together confirms that OB isolated by described method is relevant model of normal osteogenic differentiation. It is known that differentiation processes are cell and tissue specific. Difference in osteogenic differentiation between smooth muscle cells and osteoblasts was shown in murine cells (Patel et al. 2019). Osteogenic differentiation of MSC of non-bone origin is actually rather related to pathological than to physiological processes which take place during bone formation. In our previous publications we have shown that osteogenic differentiation of different types of mesenchymal cells demonstrate some variability in response to osteogenic stimuli (Aleksandra Kostina et al. 2021). In vascular and valvular calcification a response visible by expression of osteomarkers to osteogenic stimuli may vary not only in different cell types, but also might be dependent  on the type of pathological process.  (Ignatieva et al. 2017), (A. Kostina et al. 2018). Moreover, it is known that the mechanisms of osteogenic differentiation vary between species and a response of osteoblasts of non-human origin to pharmacological agents may differ from that of human osteoblasts. In our recent review and one more work it is discussed that many alterations are described for expression of transcription factor Runx2 in response to osteogenic stimuli and thus the expression of Runx2 might be cell type/model-dependent. For example, dexamethasone stimulated Runx2 production in human osteoblast cell lines, but inhibited Runx2 in rodent osteoblasts (Lobov and Malashicheva 2022). That is the reason to test antiosteoporosis drugs and materials for bone prothesis on human osteoblasts of bone origin in preclinical studies in addition to animal models. Novelty of this research is that we suggest a panel of reliable osteomarkers that match normal osteogenic changes that could be used as a reference for studying pathological calcification, bone formation and osteoporosis. We have shown that donor to donor effect is quite prominent and the level of production of osteomarkers vary both in normal conditions and after osteogenic stimulation. We suggest to pay attention to this fact while planning experiments with osteoblasts. We have shown that 96 hours is enough for detecting osteogenic changes at the mRNA level and consider this timepoint as a reliable for estimation of osteogenic changes.

References

AO Research Institute Davos, Clavadelerstrasse 8, CH-7270 Davos Platz, Switzerland, M Bruderer, Rg Richards, M Alini, and Mj Stoddart. 2014. ‘Role and Regulation of RUNX2 in Osteogenesis’. European Cells and Materials 28 (October): 269–86. https://doi.org/10.22203/eCM.v028a19.

Bernhardt, Anne, Sophie Wolf, Emilia Weiser, Corina Vater, and Michael Gelinsky. 2020. ‘An Improved Method to Isolate Primary Human Osteocytes from Bone’. Biomedical Engineering / Biomedizinische Technik 65 (1): 107–11. https://doi.org/10.1515/bmt-2018-0185.

Fujita, K., M. M. Roforth, E. J. Atkinson, J. M. Peterson, M. T. Drake, L. K. McCready, J. N. Farr, D. G. Monroe, and S. Khosla. 2014. ‘Isolation and Characterization of Human Osteoblasts from Needle Biopsies without in Vitro Culture’. Osteoporosis International 25 (3): 887–95. https://doi.org/10.1007/s00198-013-2529-9.

Ignatieva, E., D. Kostina, O. Irtyuga, V. Uspensky, A. Golovkin, N. Gavriliuk, O. Moiseeva, A. Kostareva, and A. Malashicheva. 2017. ‘Mechanisms of Smooth Muscle Cell Differentiation Are Distinctly Altered in Thoracic Aortic Aneurysms Associated with Bicuspid or Tricuspid Aortic Valves’. Frontiers in Physiology 8 (JUL): 1–12. https://doi.org/10.3389/fphys.2017.00536.

Kostina, A., A. Shishkova, E. Ignatieva, O. Irtyuga, M. Bogdanova, K. Levchuk, A. Golovkin, et al. 2018. ‘Different Notch Signaling in Cells from Calcified Bicuspid and Tricuspid Aortic Valves’. Journal of Molecular and Cellular Cardiology 114 (January): 211–19. https://doi.org/10.1016/J.YJMCC.2017.11.009.

Kostina, Aleksandra, Arseniy Lobov, Daria Semenova, Artem Kiselev, Polina Klausen, and Anna Malashicheva. 2021. ‘Context-Specific Osteogenic Potential of Mesenchymal Stem Cells’. Biomedicines 9 (6): 673. https://doi.org/10.3390/biomedicines9060673.

Lobov, Arseniy, and Anna Malashicheva. 2022. ‘Osteogenic Differentiation: A Universal Cell Program of Heterogeneous Mesenchymal Cells or a Similar Extracellular Matrix Mineralizing Phenotype?’ Biological Communications 67 (1). https://doi.org/10.21638/spbu03.2022.104.

Patel, Jessal J., Lucie E. Bourne, Bethan K. Davies, Timothy R. Arnett, Vicky E. MacRae, Caroline PD. Wheeler-Jones, and Isabel R. Orriss. 2019. ‘Differing Calcification Processes in Cultured Vascular Smooth Muscle Cells and Osteoblasts’. Experimental Cell Research 380 (1): 100–113. https://doi.org/10.1016/j.yexcr.2019.04.020.

Rickard, David J., Moustapha Kassem, Theresa E. Hefferan, Gobinda Sarkar, Thomas C. Spelsberg, and B. Lawrence Riggs. 2009. ‘Isolation and Characterization of Osteoblast Precursor Cells from Human Bone Marrow’. Journal of Bone and Mineral Research 11 (3): 312–24. https://doi.org/10.1002/jbmr.5650110305.

Yagiela, John A., and Dixon M. Woodbury. 1977. ‘Enzymatic Isolation of Osteoblasts from Fetal Rat Calvaria’. The Anatomical Record 188 (3): 287–305. https://doi.org/10.1002/ar.1091880303.

Round 2

Reviewer 2 Report

The authors have clarified and responded to most of the reviewer concerns however major editing for proper use of the English language is indispensable.

There are several deficiencies, mistakes, omitted words and improper use of words (e.g solve collagenase line84 instead dissolve; cell lines are also widely spread as a research model line43; Not compromising the importance of other cell models, we suppose that explant culture is the most relevant model of adult bone lines 43-45; Bone-marrow derived cells contain not only osteoblasts precursors, but could differentiate to other cell types lines 47-48; Approach number of cells are line170; According to flow cytometry characteristics line290; osteoclast isolation from the same source as ours (human femur) has been described [19] line309.……..and many more) throughout the entire manuscript that make it difficult to fully evaluate the scientific contribution of the article. 

Author Response

We have proofread the manuscript with the haelp of an English native speaker and corrected the text.

Reviewer 3 Report

I have no more comments for the authors.

Author Response

We have proofread the manuscript with the help of an English native speaker and corrected the text.

Reviewer 4 Report

The authors have improved the manuscript by adequately addressing the reviewers' critiques.

There are still minor mistakes in the English language:

homogeneous, not homogenic;

reproducible, not reproduceable.

Author Response

(The authors gave the same response as above.)
